# Evaluation of MoveSmart MS - an online structured exercise, Social Cognitive Theory-based behavioural coaching and peer support programme - on anxiety in multiple sclerosis

Austin Fahy[1,2], Susan Coote[3,4,5], Rebecca Maguire[1,6]*

**1** Department of Psychology, Maynooth University, Maynooth, Ireland, **2** SIM Centre for Simulation Education and Research, Royal College of Surgeon's Ireland, Dublin, Ireland, **3** Multiple Sclerosis Society of Ireland, Dublin, Ireland, **4** School of Allied Health, Faculty of Education and Health Sciences, University of Limerick, Limerick, Ireland, **5** Centre of Physical Activity for Health, Health Research Institute, University of Limerick, Limerick, Ireland, **6** Assisting Living and Learning Institute, Maynooth University, Maynooth, Ireland

* rebecca.maguire@mu.ie

## Abstract

### Background

Anxiety is a common experience among people with multiple sclerosis (PwMS). While it is known that engaging in exercise has a number of benefits, the role played by exercise in reducing anxiety has received less attention in this context. Understanding the effect of exercise on anxiety, the predictors of this change. and participant experiences will help to further develop programmes which can support PwMS.

### Aim

This study used a pre/post evaluation design (1) to assess the efficacy of a structured physiotherapist-led online exercise programme (MoveSmart) on anxiety in PwMS living in Ireland, (2) to identify the role of sociodemographic, health and psychological factors in predicting baseline anxiety, (3) to investigate how changes in these factors predict changes anxiety following completion of the programme, and (4) to explore the experiences of participants with the programme.

### Method

Data from 284 PwMS who took part in a 10-week programme between January 2021 and June 2022 were analysed. Participants provided sociodemographic and health information at baseline and completed measures of psychological factors at baseline and again on completion of the programme. Change in anxiety scores were analysed using hierarchical regression analysis. Follow-up focus groups were conducted with

**Data availability statement:** Data cannot be shared publicly for ethical reasons. There are ethical restrictions involving the sharing of data. Given the range of demographic and health-data collected, coupled with the small number of people with MS living in Ireland, there is a risk that participants could be potentially identifiable. In addition, at the time of applications for University of Limerick Ethics Committee approval (approval code 2020-12-19), the participants did not consent to share their data publicly. Data are available from the University of Limerick (contact details below) for researchers who meet the criteria for access to confidential data. Contact details: Chairman Education and Health Sciences Research Ethics Committee EHS Faculty Office University of Limerick Email: ehsresearchethics@ul.ie Ref: 2020-12-19.

**Funding:** The author(s) received no specific funding for this work.

**Competing interests:** The authors have declared that no competing interests exist.

25 participants to explore their experiences of taking part in the programme, with data analysed using reflexive thematic analysis.

## Results

Participation in the MoveSmart programme was associated with significant reductions in anxiety. Younger age, lower disability, higher Multiple Sclerosis Impact Scale-29 physical scores and higher fatigue were significantly associated with higher baseline anxiety scores, and changes in the physical impact of MS predicted changes in anxiety scores. Key themes identified through thematic analysis included 'Benefits of peer support', 'Benefits of goal setting', 'Noticed improvements' and 'Praise for programme execution'.

## Conclusion

Participation in a physiotherapist-led structured exercise programme resulted in significant improvements in anxiety, in part through reducing the impact of physical MS symptoms. Qualitative results suggest that peer support and goal-related coaching may have contributed to reductions in anxiety.

## 1. Introduction

Multiple Sclerosis (MS) is a chronic, demyelinating, neurological condition and the most common cause of non-traumatic neurological disability in young adults [1]. In addition to physical symptomology, MS has been shown to potentially affect psychological well-being, quality of life and social functioning [2,3]. Specifically, there is a clear need for research which investigates means of supporting people with multiple sclerosis (PwMS) with the common [4] and potentially debilitating [5] experience of anxiety. A number of modifiable associates of anxiety [4,6] in PwMS have been identified which may have value for use in targeted interventions. Notably, exercise activity has been shown to predict anxiety in PwMS (higher exercise activity associated with lower anxiety) [7,8] which is consistent with findings from general populations [9]. Several interventions based on exercise activity in PwMS show promising efficacy in improving a number of outcomes for PwMS including anxiety reduction [8,10–12], however, findings here are mixed. While some exercise programmes have been associated with anxiety reducing effects, others have found no such effect on anxiety [7]. This may be due to the considerable variability in recruiting methods, delivery of the interventions (e.g., online vs. in-person, group vs. individually delivered), measures of anxiety used and sample characteristics.

A recent review investigating the potential impact of exercise interventions on anxiety in PwMS highlighted the paucity of research in this area and the urgent need for more systematic evaluations of exercise interventions for anxiety in PwMS [13]. In other chronic health conditions, such as fibromyalgia and coronary heart disease [14] as well as in general populations [15] exercise has been shown to reduce symptoms

of anxiety. However, given the nature of MS-specific considerations related to anxiety, the transferability of these findings to MS populations as well as the key considerations related to tailoring these interventions for MS populations are unclear. There is a therefore a need for a better understanding of the relationship between exercise and anxiety in PwMS as well as the potential utility of exercise as a target intervention for improving anxious symptomology.

One potential mechanism through which exercise may impact anxiety is through self-efficacy, a positive construct which describes one's perception of one's own abilities [16]. In some theoretical models of anxiety, including the Intolerance Uncertainty Model (IUM) and the Contrast Avoidance Model (CAM), it is suggested that self-efficacy influences anxiety through negative problem orientation, a concept associated with increased anxiety, where higher self-efficacy (greater confidence in one's abilities) predicts lower negative problem orientation [17]. Additionally, self-efficacy is a core concept in Social Cognitive Theory (SCT), with evidence suggesting that self-efficacy (as well as other concepts core to SCT such as goal setting) is associated with physical activity in PwMS [18,19]. It is worth noting that these models of anxiety do not account for MS-specific considerations (including those related to the uncertainty of MS disease course) and thus the utility of using these models to improve understanding of the experience of anxiety in PwMS remains unclear. Despite these concerns, in a review investigating future-oriented cognitions in PwMS it was found that self- efficacy was the strongest predictor of quality of life (with higher self-efficacy predicting higher quality of life), suggesting that improvements in self-efficacy could have a number of benefits for PwMS [20]. Exercise programmes which include SCT educational components have previously been shown to have efficacy in improving walking related outcomes in PwMS, however the potential impact of these programmes on anxiety, while promising, is less clear [12,21,22]. In an Irish context, one study found that self-efficacy significantly predicted the variance in physical activity scores, with anxiety at lower or mean levels significantly moderating the relationship between self-efficacy and physical activity [23]. This suggests exercise may have particular benefit for those with less severe anxiety symptoms. Given research showing that exercise can increase self-efficacy in PwMS [24], as well as evidence associating increased self-efficacy with lower anxiety in PwMS [4,6], exercise programmes that increase self- efficacy may be particularly beneficial in reducing anxiety in PwMS.

It should be noted that promotion of physical activity in PwMS should be done with care. While exercise behaviours can provide PwMS with a sense of control over certain symptoms, poor adherence to prescribed exercise can create feelings of guilt and worry that disease progression is the 'fault' of the PwMS [25]. While modifiable psychosocial constructs like self-efficacy play an important role in exercise outcomes, evidence suggests that MS symptoms such as fatigue and walk-ing limitations coupled with these psychosocial constructs explain more of the variance in physical activity than psycho-social constructs alone [26]. Additionally, meta-regression analysis of exercise programmes for PwMS found significantly larger antidepressant effects of the programmes in which there were significant improvements to fatigue symptoms [27]. These findings highlight the importance of targeting both physical and psychosocial factors in the promotion of physical activity for PwMS.

MS Ireland is a charitable organisation which plays a crucial role in the provision of information and support for PwMS in Ireland [28]. MS Ireland recently developed a suite of structured exercise programmes (collectively termed the "MoveSmart" programme) designed for different cohorts of PwMS (MS Ireland, n.d.). This physiotherapist-led programme is conducted over a 10-week period and features symptom-focused exercise, behavioural coaching based on the prin-ciples of SCT and peer support for PwMS. This programme builds on a successful RCT of the "Step It Up" programme which also combined exercise and SCT based behavioural coaching in PwMS with low to moderate disability. The Step It Up RCT was shown to have positive impacts on walking related outcomes [21,22], as well as anxiety, depression and on the physical impact of MS [29]. Furthermore, qualitative evaluations of this programme highlighted potential psychological benefits, including reduced anxiety and improved self-efficacy [30]. In addition, the peer support experienced during the programme was highlighted as beneficial, with suggestions that peer support elements were as important to participants as planned exercise activities [30]. Given findings highlighting the psychological benefits of peer support [31] and that lev-els of need for social support are high [32], there may be particular value in having this as an element of this programme.

It should be noted that while these results are promising, anxiety was not the primary outcome assessed in any of the Step It Up studies, and furthermore, the mechanisms for impacting change in anxiety were not investigated. Furthermore, MoveSmart builds upon the Step It Up programme through the adoption of symptom-tailored principles exemplified by the addition of MS symptom-specific modules alongside the embedding of core modules from Step It Up.

This study aimed to 1) explore if and how the experience of anxiety changed among PwMS who engaged in the MoveSmart exercise programmes, as well as 2) identifying any sociodemographic, health, psychological or exercise related associates of anxiety at baseline. Additionally, this study aimed to 3) explore the likely mechanisms behind any change in anxiety and 4) explore participant experiences with the MoveSmart programmes.

## 2. Method

### 2.1 Data sources

This study involves analysis of data from participants who took part in the MoveSmart MS programme. Participants were provided an information sheet with programme details and required to provide written consent before taking part in the programme. The MoveSmart MS programme consisted of 10-weeks of group-based behavioural coaching with one online session per week. Sessions included covering SCT-based outcome expectancies, self-efficacy, goal setting and adjustment and symptom management and with the content of the sessions dependent on the disability level of the group. For example, while all participants engaged in core modules which were derived from the Step It Up programme (e.g., outcome expectations, action planning etc.) participants also engaged in symptom-specific modules (e.g., dropfoot and walking, falls prevention programme). These symptom-specific modules were not featured in the Step It Up programme and represent a practical and conceptual iteration. Each participant had 90 minutes of contact with a physiotherapist each week consisting of 45 minutes of exercise-based activity and 45 minutes of behavioural coaching. Participants were asked to complete a survey which included questions on demographics (age and sex), health information including type of MS and the Patient Determined Disease Steps Scale (PDSS) [33,35], as well as validated measures relating to psychological well-being, lifestyle and disease-related variables both pre and post their participation in the MoveSmart MS programme. More detail on the specific measures used is described in section 2.2. Qualitative data was collected by researchers not involved in programme delivery through three focus groups conducted on the 5th, 19th and 20th of August 2021. Questions explored participant experiences with the programmes delivery and content, as well as any criticisms or benefits derived from the programme. All participants in the MoveSmart programme between January 1st 2021 and June 30th 2022 received an email invitation to be involved with focus group qualitative data collection, with all those who accepted the invitation included. Ethical approval for this study was granted by the University of Limerick ethics committee (REF No. 2020-12-19) on December 19th 2020, with permission for the authors to conduct analysis on the anonymised dataset and transcribed focus group data on July 19th 2023.

### 2.2 Measures

**2.2.1 Anxiety.** Anxiety was measured using the State-Trait Anxiety Inventory trait-subscale (STAI-T) [34]. This measure consists of 20 items (e.g., "I worry too much over something that really doesn't matter"), with answers logged on a 4-point Likert scale (e.g., from "Almost Never" to "Almost Always"). Total STAI-T scores range from 0–60 with higher scores indicative of higher level of trait anxiety. The STAI is a commonly used means of evaluating anxiety in PwMS [6].

**2.2.2 Mobility disability.** Patient reported mobility disability was assessed using the Patient Determined Disease Steps scale (PDDS) [33,35]. This measure consists of nine items with participants selecting the item that best describes their current level of disability. Items are scored as follows: 0 – normal; 1 – mild disability; 2 – moderate disability; 3 – gait disability; 4 – early cane; 5 – late cane; 6 – bilateral support; 7 – wheelchair/scooter; and 8 – bedridden, with higher

scores indicative of higher levels of disability. A recent systematic review [35] found good validity and test re-test reliability with recommendations for use in samples of PwMS with mild to moderate disability.

**2.2.3 Exercise self-efficacy.** The Exercise Self-Efficacy Scale (EXSE) was used to assess participant beliefs about their ability to exercise at moderate intensity, three times per week, for a minimum of 40 minutes per session [36]. The six-item version of the EXSE asks participants to indicate their confidence in maintaining this level of exercise for 1, 2, 3, 4, 5, and 6 weeks with each time period represented by a single item. logged using an 11-point Likert scale ranging from 0% to 100% in 10% intervals. These scores are then converted and averaged with total scores ranging from 0–100. The EXSE has been used frequently to evaluate exercise self-efficacy in MS populations, with one study [37] reporting an estimated internal consistency of 0.99.

**2.2.4 Fatigue.** The Modified Fatigue Impact Scale (MFIS) was used to assess levels of fatigue experienced by participants in this sample [38]. The full 21- item version was used with dimensions of physical, cognitive and psychosocial fatigue assessed (e.g., "I have been less able to complete tasks that require physical effort"). Responses are indicated using a 5-point Likert scale (never, rarely, sometimes, often, almost always). Total scores range from 0–84 with higher scores indicative of higher levels of fatigue. The MFIS has been used frequently to assess fatigue in MS populations [39], with good internal consistency (Cronbach's Alpha > 0.9) [40].

**2.2.5 Physical impact of MS.** The Multiple Sclerosis Impact Scale –29 v2 (MSIS-29) physical impact subscale was used to assess perceived physical impact of MS in this sample [41]. The physical impact subscale of the MSIS-29 contains 20 items assessing different dimensions of disability, physical symptomology and quality of life impacts in a timescale which includes the last two weeks (e.g., "In the past two weeks, how much have you been bothered by..." "Having to cut down the amount of time you spent on work or other daily activities?"). Answers are logged using a 5-point Likert scale (not at all, a little, moderately, quite a bit and extremely) where each item is scored 1–5, with total scores ranging from 0 to 100. The MSIS-29 has been used frequently in MS populations [42] and has been shown to have good reliability and sensitivity (Cronbach's Alpha > 0.8) [43].

**2.2.6 Physical activity.** The Godin Leisure Time Exercise Questionnaire (GLTEQ) was used to assess physical activity [44]. It should be noted that GLTEQ weekly leisure activity scores (GLTEQ WLA) as opposed to GLTEQ health contribution scores (GLTEQ HCS) were used. GLTEQ WLA scores account for low, moderate and high intensity exercise, while GLTEQ HCS scores account only for moderate and high intensity exercise. The decision to use GLTEQ WLA scores was taken to account for PwMS who exercise at lower levels of intensity.

**2.2.7 Walking ability.** The Multiple Sclerosis Walking Scale (MSWS) was used to assess walking ability in this sample [45]. The MSWS contains 12 items which ask participants to indicate the extent to which MS has impacted on different abilities related to standing and walking during the last two weeks (e.g., "limited your balance when standing or walking?"). Answers are logged on a 5-point Likert scale (not at all, a little, moderately, quite a lot, extremely). These scores were then standardised to range from 0–100, where higher scores indicate greater difficulty with walking. The MSWS has shown good reliability for use in MS populations [46].

**2.2.8 Additional measures.** Participants also completed the Quick Inventory of Depressive Symptomology (QIDS) [47] and the psychological subscale of the MSIS-29 [41] as part of the pre vs post-test battery, however these factors were considered outside the focus of the current study due to conceptual overlap between these factors and anxiety and issues with multicollinearity.

## 2.3 Data analysis

A mixed-methods approach was used for data analysis. Descriptive statistics were calculated with means, ranges and standard deviations presented for continuous variables and frequencies calculated for categorical variables. All categorical variables were subsequently binary coded prior to analysis. For example, MS type was categorised into RRMS (Relapsing Remitting MS) or progressive MS (including Secondary Progressive MS (SPMS) or Primary Progressive MS (PPMS)).

To address this study's first aim, a paired sample t-test of pre vs. post intervention anxiety scores was conducted. After conducting correlational analyses to assess assumptions of multicollinearity, linearity and homoscedasticity, hierarchical regression modelling was used to assess associations between (1) sociodemographic factors (age, sex), (2) MS characteristics (MS type, time since diagnosis, PDDS), (3) subjective health measures (EXSE, MFIS, MSIS-29 physical subscale), (4) physical activity (GLTEQ) and anxiety at baseline. Power calculations were conducted using G*Power software [48] to determine required sample size for hierarchical regression. The statistical test used was "Linear multiple regression: Fixed model, R squared deviation from zero". A medium effect size ($f^2 = 0.15$, corresponding to $R^2 = 0.13$,) was assumed, with an alpha level of 0.05 and a desired power of 0.95. The analysis indicated that a minimum of 166 participants were needed (with 9 predictors).

A separate hierarchical regression model was run using the same sociodemographic factors collected at baseline, and the pre vs. post-test changes in health status, MS-related factors and GLTEQ scores in their extent to which they could predict changes in anxiety (post-pre programme anxiety scores).

Furthermore, a sensitivity analysis comparing baseline data from completers vs non-completers was conducted to assess any potential differences between these groups on the measures employed. Additionally, STAI scores were analysed in accordance with the guidelines to allow for the description of the clinical significance of our sample's anxiety scores. This was done to allow for a clear description of anxiety scores and to facilitate potential comparisons with samples using other anxiety measures.

Finally, reflexive thematic analysis, based on the principles of Braun & Clarke [49], was used to analyse qualitative data collected from focus groups about participants' experiences of the MoveSmart programme. All participants were invited to take part in focus groups via email. Analysis of focus group data began with an initial period of the researchers (AF and RM) familiarising themselves with the data, after which initial codes were constructed by AF using MAXQDA 2020. AF used these initial codes to create initial themes, which were then presented to RM. These initial theme names and definitions were further refined through review and discussion until the point of data saturation, the point at which no new codes or themes could be identified. At this point AF created the final report based on these discussions.

Narrative integration was used to integrate qualitative and quantitative findings [50]. Specifically, a weaving approach was chosen given its utility for research where qualitative and quantitative results are thematically connected and where both types of data 'weave' back and forth around similar concepts. This process involves reporting findings theme by theme or concept by concept, describing findings from both quantitative and qualitative methods in the same section [51].

## 3. Results

### 3.1 Sample characteristics

294 participants completed the MoveSmart MS exercise programme pre-test questionnaire. 170 participants completed post-programme questionnaire, however, 10 of these were excluded from analysis due to missing pre-programme data, resulting in a 54.4% attrition rate for the post-intervention survey and a total of 160 participants who completed both pre and post programme measures. All participants were invited to participate in a follow-up focus group of which a small selection of participants (n = 25) accepted. A single 30-minute focus group was conducted with this pool of participants. The focus group participants consisted of 8 men and 17 women, 20 of whom had a diagnosis of RRMS, 4 with a diagnosis of SPMS and one participant with PPMS. In addition, the mean age of focus group participants was 55.6 years, with a mean time since diagnosis of 16 years.

Tables 1 and 2 include descriptive statistics for this sample. In line with population norms, the sample was mostly female (82.7%) with an age range of 24–88 years (mean age = 50.37, SD = 11.75). Time since diagnosis ranged from less than a year to 53 years (mean = 12.45 years, SD = 10.51). In terms of MS type, most participants (64.8%) reported having RRMS, followed by SPMS (15.8%) and PPMS (14.4%).

**Table 1. Sample demographics for categorical variables.**

| Variable | N | % |
|---|---|---|
| **Sex** | | |
| Male | 49 | 17.30% |
| Female | 235 | 82.70% |
| **MS type** | | |
| Relapsing Remitting MS | 184 | 64.80% |
| Secondary Progressive MS | 45 | 15.80% |
| Primary Progressive MS | 41 | 14.40% |
| Other | 14 | 4.90% |
| Missing | – | – |

**Table 2. Comparison of pre and post study measures.**

| Variable | Baseline (T1) | | | | | Post-test (T2) | | | | | Change from T1 to T2 | | | |
|---|---|---|---|---|---|---|---|---|---|---|---|---|---|---|
| | N | Missing | M | SD | Range | N | Missing | M | SD | Range | M | t | Cohen's d | p |
| STAI | 284 | – | 20.36 | 11.26 | 0-54 | 160 | 124 (43.7%) | 17.73 | 10.27 | 0-46 | −2.90 | 4.926 | −.39 | <.001 |
| Age | 283 | 1 (0.4%) | 50.37 | 11.75 | 24-88 | – | – | – | – | – | – | – | | – |
| Time Since Dx | 284 | – | 12.45 | 10.51 | 0-53 | – | – | – | – | – | – | – | | – |
| PDDS | 282 | 2 (0.7%) | 3.00 | 2.16 | 0-7 | – | – | – | – | – | – | – | | – |
| MFIS | 284 | – | 43.51 | 14.95 | 0-84 | 160 | 124 (43.7%) | 36.12 | 13.07 | 7-72 | −7.99 | 8.785 | .52 | <.001 |
| EXSE | 284 | – | 68.02 | 30.92 | 0-100 | 160 | 124 (43.7%) | 72.76 | 26.82 | 0-100 | 3.58 | −1.588 | .09 | >.05 |
| MSIS-29 physical | 277 | 7 (2.5%) | 39.48 | 22.73 | 0-98.33 | 156 | 128 (45.1%) | 31.36 | 20.27 | 0-90 | −9.38 | 8.327 | .50 | <.001 |
| GLTEQ-WLA | 284 | – | 20.12 | 18.80 | 0-110 | 158 | 126 (44.4%) | 29.18 | 19.59 | 0-101 | 9.11 | −6.564 | −.039 | <.001 |
| MSWS | 284 | – | 49.75 | 32.24 | 0-100 | 160 | 124 (43.7%) | 44.97 | 31.15 | 0-100 | −7.93 | 6.104 | .36 | <.001 |

EXSE = Exercise Self-Efficacy Scale; GLTEQ-WLA = Godin Leisure Time Exercise Questionnaire Weekly Leisure Activity; MFIS = Modified Fatigue Impact Scale; MSIS-29 physical = Multiple Sclerosis Impact Scale physical subscale; MSWS = Multiple Sclerosis Walking Scale; PDDS = Patient-Determined Disease Steps; STAI = State-Trait Anxiety Inventory.

After adjusting STAI scores in accordance with the guidelines using the one standard deviation non-gendered clinical cut off (STAI = 32.59), 20.4% of the sample scored above the cut off for clinical levels of anxiety at baseline. Using the same cut-off, only 6.25% of participants' scores indicated clinical levels of anxiety post-programme.

### 3.2 Sensitivity analysis

An independent samples two-tailed t-test was used to assess any potential differences between the 160 participants who completed post-test measures and the 124 participants who completed only pre-test measures. There were no significant differences at baseline in any of the measures used in this analysis, but non-completers (M = 48.05, SD = 11.91) were significantly younger on average, $t(281) = 2.95$, $p = .003$, than completers (M = 52.15, SD = 11.35) with a shorter time since diagnosis, $t(282) = 2.15$, $p = .033$ for non- completers (M = 10.94, SD = 10.5) than completers (M = 13.63, SD = 10.39).

### 3.3 Comparison of pre and post programme measures

Paired samples t-tests results (see Table 2) indicated that pre-programme (baseline) STAI scores ($M = 20.69$, $SD = 10.85$) were significantly higher than post-programme STAI scores ($M = 17.79$, $SD = 10.18$), $t(159) = 4.926$, $p < .001$. In addition, pre-programme GLTEQ scores ($M = 16.99$, $SD = 1.35$) were significantly lower than post-programme GLTEQ scores ($M = 19.59$, $SD = 1.558$), $t(157) = -6.564$, $p < .001$. Furthermore, pre-programme MFIS scores ($M = 44.11$, $SD = 15.17$) were significantly higher than post-programme MFIS scores ($M = 36.12$, $SD = 13.07$), $t(159) = 8.785$, $p < .001$. Pre-programme MSIS-29 psychological scores ($M = 38.92$, $SD = 21.66$) were significantly higher than post-programme MSIS-29 psychological scores ($M = 28.37$, $SD = 18.29$), $t(158) = 7.343$, $p < .001$. Similarly, pre-programme MSIS-29 physical scores ($M = 40.74$, $SD = 21.42$) were significantly higher than post-programme MSIS-29 physical scores ($M = 31.37$, $SD = 20.34$), $t(154) = 8.327$, $p < .001$. Pre-programme MSWS scores ($M = 52.89$, $SD = 30.28$) were also significantly higher than post-programme MSWS scores ($M = 44.97$ $SD = 31.15$), $t(160) = 6.104$, $p < .001$. Interestingly, no statistically significant difference was found between pre-programme EXSE scores ($M = 69.19$, $SD = 30.04$) and post-programme EXSE scores ($M = 72.76$, $SD = 26.82$), $t(159) = -1.588$, $p > .05$.

### 3.4 Predictors of anxiety

Correlational analysis assessing the relationship between pre-test (baseline) STAI scores and the predictor variables revealed significant multicollinearity between MSWS and PDSS ($r = .849$), and between MSWS and MSIS physical scores ($r = .788$). Variation inflation factors (VIF) and tolerance scores were also consulted with VIFs greater than 4 and tolerance values less than 0.2 considered indicative of multicollinearity. As a result, MSWS scores were not included in the hierarchical regression analysis. No outliers were removed from the data set. An alpha level of .05 was used for this analysis.

The first hierarchical regression analysis assessed the relationship between pre-programme (baseline) STAI scores and four blocks of predictor variables. Block 1 consisted of sociodemographic factors, specifically age and gender. Block 2 consisted of MS characteristics, including MS type (dichotomised into RRMS vs progressive MS (SPMS and PPMS)), time since diagnosis (five years or less vs greater than five years), and PDDS scores. Block 3 consisted of self-reported subjective health measures including MFIS scores, EXSE scores, and MSIS-29 physical subscale scores. Block 4 consisted of levels of physical activity specifically GLTEQ-WLA scores.

The first regression analysis predicted 30.5% of the variance in pre- programme STAI scores (see Table 4). Block 1 ($F(2, 257) = 8.81$; $p < .001$) and Block 3 ($F(8, 251) = 13.79$; $p < .001$) significantly contributed to the model. In the final model age ($\beta = -.186$, $p = .003$), PDDS ($\beta = -.349$, $p < .001$), MFIS ($\beta = .168$, $p = .039$), and MSIS-29 physical impact ($\beta = 0.464$, $p < .001$) were significant predictors of pre-test STAI scores. Specifically, higher mobility disability, higher physical impact of MS, higher fatigue and younger age were significantly associated with increased STAI scores. Overall, the model was significant ($F(9, 250) = 12.21$; $p < .001$). Table 3 and 4 display the results of analyses in relation to pre-programme STAI scores.

A separate hierarchical regression was used to investigate change in STAI scores following completion of the programme using the four blocks of predictor variables. Results of correlational analyses used to assess the relationship between change STAI scores and the predictor variables are presented in Table 5. Block 1 and 2 consisted of the same variables used in previous models. Block 3 consisted of (post-pre programme) MFIS change scores, EXSE change scores, and MSIS-29 physical subscale change scores. Block 4 consisted of GLTEQ-WLA change scores.

This analysis predicted 24.2% of the variance in STAI change scores (see Table 6). Only Block 3 ($F(8, 135) = 5.40$; $p < .001$) significantly contributed to the model. In the final model, change in MSIS-29 physical impact ($\beta = 0.372$, $p = .001$) was the only significant predictor of change in STAI scores. Specifically, reductions in the physical impact of MS were significantly associated with decreased STAI scores. Overall, the model was significant ($F(9, 134) = 4.76$; $p < .001$).

**Table 3. Correlation matrix for hierarchical regression of baseline STAI scores.**

| Variable | n | M | SD | 1 | 2 | 3 | 4 | 5 | 6 | 7 | 8 | 9 | 10 |
|---|---|---|---|---|---|---|---|---|---|---|---|---|---|
| 1. Baseline STAI | 284 | 21.45 | 11.14 | — | | | | | | | | | |
| 2. Age | 283 | 50.37 | 11.75 | −.28** | — | | | | | | | | |
| 3. Gender | 284 | 1.83 | .38 | .05 | −.12* | — | | | | | | | |
| 4. MS Type [1=RRMS, 2=SPMS or PPMS] | 270 | 1.47 | .74 | −.15* | .38** | −.30** | — | | | | | | |
| 5. Time since diagnosis [1=5 years or less, 2=longer than 5 years] | 284 | 1.67 | .47 | −.20** | .39** | −.07 | .16** | — | | | | | |
| 6. PDDS | 282 | 3.00 | 2.16 | −.13* | .45** | −.21** | .57** | .40** | — | | | | |
| 7. Baseline MFIS | 284 | 43.51 | 14.95 | .35** | .08 | −.05 | .13* | .05 | −.30** | — | | | |
| 8. baseline EXSE | 284 | 68.02 | 30.92 | −.14* | −.04 | .09 | −.04 | −.09 | −.15* | −.26** | — | | |
| 9. Baseline MSIS-29 physical subscale | 277 | 39.48 | 22.73 | .29** | .23** | −.19** | .36** | .23** | .65** | .69** | −.27** | — | |
| 10. Baseline GLTEQ- WLA | 284 | 20.12 | 18.80 | −.03 | −.14* | −.06 | −.08 | −.18** | −.23** | −.28 | .30** | −.26** | — |

*p<.05; ** p<.01; EXSE=Exercise Self-Efficacy Scale; GLTEQ-WLA=Godin Leisure Time Exercise Questionnaire Weekly Leisure Activity; MFIS=Modified Fatigue Impact Scale; MSIS-29 physical=Multiple Sclerosis Impact Scale physical subscale; PDDS=Patient-Determined Disease Steps; Primary Progressive Multiple Sclerosis=PPMS; RRMS=Relapsing Remitting Multiple Sclerosis; SPMS=Secondary Progressive Multiple Sclerosis; STAI=State-Trait Anxiety Inventory.

**Table 4. Hierarchical regression analysis investigating predictors of pre-programme (baseline) STAI scores.**

| | B | 95% CI | | SE B | β | R² | ΔR² |
|---|---|---|---|---|---|---|---|
| | | LL | UL | | | | |
| **Step 1: Sociodemographic factors** | | | | | | .064 | .064** |
| Constant | 31.349 | 22.328 | 40.369 | 4.581 | | | |
| Age | −.173* | −.287 | −.059 | .058 | −.186* | | |
| Sex [1=male, 2=female] | 1.268 | −1.874 | 4.410 | 1.595 | .045 | | |
| **Step 2: MS Characteristics** | | | | | | .07 | .006 |
| Constant | 33.929 | 23.967 | 43.891 | 5.059 | | | |
| MS Type [1=RRMS, 2=SPMS or PPMS] | −.497 | −2.461 | 1.467 | .997 | −.034 | | |
| Time since diagnosis [1=5 years or less, 2=longer than 5 years] | −1.274 | −4.067 | 1.518 | 1.418 | −.054 | | |
| PDDS | −1.774* | −2.659 | −.888 | .450 | −.349*** | | |
| **Step 3: Subjective Health** | | | | | | .305 | .236** |
| Constant | 23.500 | 13.589 | 33.411 | 5.032 | | | |
| Baseline EXSE | −.055 | −.059 | .020 | .020 | −.055 | | |
| Baseline MFIS | .168* | .006 | .237 | .059 | .168* | | |
| Baseline MSIS-29 physical impact subscale | .464 | .127 | .315 | .048 | .464 | | |
| **Step 4: Exercise Habits** | | | | | | .305 | .000 |
| Constant | 23.316 | 13.104 | 33.528 | 5.185 | | | |
| Pre-GLTEQ WLA | .009 | −.061 | .071 | .849 | .009 | | |

*p<.05; **p<.01; ***p<.001; CI=confidence interval; EXSE=Exercise Self-Efficacy Scale; GLTEQ-WLA=Godin Leisure Time Exercise Questionnaire Weekly Leisure Activity LL=lower limit; MFIS=Modified Fatigue Impact Scale; MSIS-29 physical=Multiple Sclerosis Impact Scale physical subscale; PDDS=Patient-Determined Disease Steps; Primary Progressive Multiple Sclerosis=PPMS; RRMS=Relapsing Remitting Multiple Sclerosis; SPMS=Secondary Progressive Multiple Sclerosis; STAI=State-Trait Anxiety Inventory; UL=upper limit.

**Table 5. Correlation matrix for hierarchical regression of STAI change scores.**

| Variable | n | M | SD | 1 | 2 | 3 | 4 | 5 | 6 | 7 | 8 | 9 | 10 |
|---|---|---|---|---|---|---|---|---|---|---|---|---|---|
| 1.STAI change | 160 | −2.9 | 7.45 | — | | | | | | | | | |
| 2. Age | 283 | 50.37 | 11.75 | .17** | — | | | | | | | | |
| 3. Gender | 284 | 1.83 | .38 | −.07 | −.12* | — | | | | | | | |
| 4. MS Type [1 = RRMS, 2 = SPMS or PPMS] | 270 | 1.47 | .74 | .11 | .38** | −.30** | — | | | | | | |
| 5. Time since diagnosis [1 = 5 years or less, 2 = longer than 5 years] | 284 | 1.67 | .47 | .19* | .39** | −.07 | .16** | — | | | | | |
| 6. PDDS | 282 | 3.00 | 2.16 | .04 | .45** | −.21** | .57** | .40** | — | | | | |
| 7. MFIS change | 160 | −7.99 | 11.50 | .37** | .13 | −.07 | .15 | .11 | .07 | — | | | |
| 8. EXSE change | 160 | 3.58 | 28.49 | −.11 | −.11 | .13 | −.12 | −.12 | −.13 | −.24** | — | | |
| 9. MSIS-29 physical subscale change | 158 | −9.38 | 14.02 | .48** | .12 | −.09 | .12 | .19* | .01 | .69** | .19* | — | |
| 10. GLTEQ-WLA change | 155 | 9.11 | 17.44 | −.16* | −.08 | −.02 | −.09 | −.12 | −.07 | −.20* | .08 | −.30** | — |

*p < .05; ** p < .01; EXSE = Exercise Self-Efficacy Scale; GLTEQ-WLA = Godin Leisure Time Exercise Questionnaire Weekly Leisure Activity; MFIS = Modified Fatigue Impact Scale; MSIS-29 physical = Multiple Sclerosis Impact Scale physical subscale; PDDS = Patient-Determined Disease Steps; Primary Progressive Multiple Sclerosis = PPMS; RRMS = Relapsing Remitting Multiple Sclerosis; SPMS = Secondary Progressive Multiple Sclerosis; STAI = State-Trait Anxiety Inventory.

**Table 6. Hierarchical regression analysis investigating predictors of change STAI scores.**

| | B | 95% CI LL | 95% CI UL | SE B | β | R² | ΔR² |
|---|---|---|---|---|---|---|---|
| **Step 1: Sociodemographic factors** | | | | | | .037 | .037 |
| Constant | −6.866 | −15.128 | 1.396 | 4.179 | | | |
| Age | .075 | −.030 | .181 | .053 | .117 | | |
| Gender [1 = male, 2 = female] | −.173 | −3.118 | 2.772 | 1.498 | −.010 | | |
| **Step 2: MS Characteristics** | | | | | | .078 | .041 |
| Constant | −12.321 | −21.818 | −2.824 | 4.803 | | | |
| MS Type [1 = RRMS, 2 = SPMS or PPMS] | .594 | −1.169 | 2.356 | .891 | .066 | | |
| Time since diagnosis [1 = 5 years or less, 2 = longer than 5 years] | 2.061 | −.669 | 4.791 | 1.380 | −.054 | | |
| PDDS | −.298 | −1.008 | .412 | .359 | .124 | | |
| **Step 3: Subjective Health Measures** | | | | | | .242 | .164** |
| Constant | −8.306 | −17.157 | .545 | 4.476 | −.084 | | |
| EXSE change | −.001 | −.040 | .039 | .020 | −.003 | | |
| MFIS change | .037 | −.098 | .171 | .068 | .058 | | |
| MSIS-29 physical impact change | .200 | .082 | .318 | .060 | .372** | | |
| **Step 4: Exercise Habits** | | | | | | .242 | .000 |
| Constant | −8.224 | −1.008 | .412 | 4.591 | | | |
| GLTEQ WLA change | −.003 | −.070 | .064 | .034 | −.007 | | |

*p < .05; **p < .01; ***p < .001; CI = confidence interval; EXSE = Exercise Self-Efficacy Scale; GLTEQ-WLA = Godin Leisure Time Exercise Questionnaire Weekly Leisure Activity; LL = lower limit; MFIS = Modified Fatigue Impact Scale;MSIS-29 physical = Multiple Sclerosis Impact Scale physical subscale; PDDS = Patient-Determined Disease; Primary Progressive Multiple Sclerosis = PPMS; RRMS = Relapsing Remitting Multiple Sclerosis; STAI = State-Trait Anxiety Inventory; SPMS = Secondary Progressive Multiple Sclerosis; UL = upper limit.

### 3.5 Reflexive thematic analysis

Transcribed data from three focus group interviews was analysed resulting in the identification of four main themes. Three of these themes, 'Benefits of peer support', 'Benefits of goal setting' and 'Noticed Improvements', focus primarily on participant outcomes and elements of the MoveSmart programme that participants felt impacted/would impact these outcomes. The final theme 'Praise for programme execution' focuses specifically on elements related to engagement in and the delivery of the MoveSmart programme. Table 7 includes additional details on main themes and subthemes.

**3.5.1 Theme 1: Benefits of peer support.** While the primary focus of the MoveSmart programme was exercise habits and symptom management, it is clear that the programme also provided an excellent source of peer support, which may potentially contribute to the observed psychological benefits. Responses indicated that participants experienced benefit from this peer support in a number of ways including a sense of comradery and understanding: *'I would value the programme for the contact with other people who are experiencing this stuff and it's for good morale I think, and it certainly would have cheered me up. Perfect.'.* Additionally, the sharing of experiences allowed for the sharing of strategies or tips from those with similar challenges, with this facilitation of open forum receiving high praise: *'through the sharing with the group you pick up ideas, you pick up possible solutions…'.*

Finally, it was apparent that those who had continued to meet after the programme had, by self-report, maintained good levels of physical activity, with participants who had not had the additional meetups noting that this had resulted in lower

**Table 7. Themes identified from focus groups.**

| Theme | Subtheme | Representative Quotes |
|---|---|---|
| Benefits of peer support | Shared experience provides comfort and knowledge | *'I also want to say that it, the, the, the people that I encountered online during the program, it was interesting to hear about their different strategies and their, their problems where, and how they got over them. So I think it's one of the best programs I've ever done.'* |
| | Desire for face-to-face | *'So, uh, the only thing I'd like add in is that maybe once a year, we all get together physically, just to say hello you know.'* |
| | Continued meet-ups encouraged continued exercise habits | *'Our group has been trying to still meet up every week and do the exercises and that's I think everybody is liking that, and like everybody doesn't manage to do it every week, but it's, it's, it's a help that we're continuing it on'* |
| Benefits of goal setting | Flexible challenges inspire action and autonomy | *'I had this whole notion of adjustable goals eh ok if you plan to do a session today and you didn't manage to it's okay. Just, reset your goals and get back to do it tomorrow.'* |
| | Routine | *'I don't know about the rest of ye that every week, you have the same sessions and every week you, um, you had, you knew the routines and I think I was great that, um, that it was consistent.'* |
| Noticed improvements | Symptoms | *'And, um, my daughter who lives with me says, um, mum, she said you've improved so much such she said since you've started that because she used to be so wobbley she said, and slow and steady, and now you've become much stronger'* |
| | Self-efficacy Understanding of MS | *'And again, It probably gave me the confidence in my body to, to adapt, you know, and to hone in on the areas to help my body function, the way it should be'*<br>*'Um, it gave me, um, a greater understanding of the, um, the different types of balance issues and dizziness, uh, people can experience with Ms. Um, I just assumed everybody who said they have balance problems like me, but actually there's lots of nuances to it. 'Um, so that gave me a greater understanding'* |
| Praise for programme execution | Online setting | *'I thought it was great eh, especially because, eh, we couldn't go out anywhere, you know, eh, and, I think as well, it's the fact that all you have to do is turn on the computer and boom you're there like, you know sometimes the logistics, well the logistics for me, for getting to a meeting. It would often be a nightmare. To find parking, where's the jacks, where am I going to have any energy, em, all that's out the window where I'm at home and you just turn on the computer, and there you are'* |
| | Programme coordinators | *'Yeah, likewise. I thought it was excellent. I really felt that I was in the hands of professionals. Uh, you know they knew what they were doing and, it worked very well.'* |

exercise levels than during the programme: *'And we set up a WhatsApp group,…, one of these days I am going to do a zoom with the lot of us and see could we do a little bit of exercise or could we do something, do I continue the exercises?.*

### 3.5.2 Theme 2: Benefits of goal setting.

In terms of the psychological impact of the programme, goals were reported as a central concept by many. Goal adjustment and reengagement in particular allowed participants to engage in goal setting while allowing for the impact of MS symptoms, a process generally described as being empowering and motivating: *'I wasn't expecting the kind of discussion about the MoveSmart object…, the smart objectives and setting objectives, and setting goals for yourself. And I found that really useful that, it translated to other elements of my life that had, say, nothing to do with the exercises. So it exceeded my expectations'.*

Additionally, the structure and regularity provided by the MoveSmart programme received considerable praise and seemed to help participants work towards their individual goals, as well as generally providing something for participants to *'look forward to'* which was reported as positively impacting morale. *'And, um, yes, certainly the routine was lovely because we were not meeting people socially, but we had our Thursday morning group and I used to look forward to meeting up with the buddies, um, from different counties'.*

### 3.5.3 Theme 3: Noticed improvements.

Participants reported that they had experienced noticeable improvements in MS-symptoms (dizziness, gait etc.) with several participants reporting that people around them (friends, family etc.) had made comments on these improvements, resulting in a positive psychological impact *'because I'm not using the stick it is quite obvious. So people have noticed, basically anyone, any of my friends um, have noticed, yeah'.*

Participants also described an improved understanding of MS in general, as well as the impacts and strategies that can be used to mitigate against the impact of symptoms: *'Yes, it did impact on my understanding of MS and the whole way I could approach things. Um, and yes, it, will change, has changed things as I go forward, because I am building in little bits of exercise every day'.* This understanding was often (though not exclusively) tied to the expression of improved self-efficacy both in terms of greater perceived control and greater perception of one's abilities: *'I think having the confidence made a difference to when I get it when, you know, if I, if I, I'd be waiting for dizziness. So do you know, it's hard to explain. I'm not doing that now. If it comes along, I can deal with it. I don't have to be nervous about it I can deal with it.'*

### 3.5.4 Theme 4: Praise for programme execution.

Responses regarding the quality of the experience of the MoveSmart programme indicated a very high level of satisfaction, with little to no consistent criticisms. The online setting received consistently high levels of praise with suggestions that it facilitated greater ease of engagement and symptom management: *'I eh thought it was very good, eh, again, eh, you know, I wasn't dependent on anyone if I had to go to outdoor I wasn't dependent on getting someone to bring me there or anything. Just so easy to just go into the room set it up and everything like. Probably, if it goes in, you know have to travel, I probably wouldn't have got to do it all'.*

Despite satisfaction with the online delivery of the programme, some participants expressed a desire for some in-person meet ups. It was suggested that these may occur irregularly and the potential to provide additional social benefits through these irregular meet ups was highlighted: *'So, uh, the only thing I'd like add in is that maybe once a year, we all get together physically, just to say hello you know.'.*

Praise for programme coordinators was also very high. Participants highlighted programme coordinators understanding, clear dissemination of information and their ability to make participants feel inspired: *'I thought it was excellent. I think it was put together with such professionalism, expertise and sensitivity. Really I've no complaints'. 'And I must say like they, um, the physios as well, like, you know, ANONYMISED?. She was very good in making sure that, um, she could see you. Do you know what I mean? And making sure that you were safe'.*

## 4. Discussion

This study highlights that the MoveSmart programme resulted in a number of benefits for PwMS, including significant reductions in anxiety. These results suggest however that this may not necessarily be due to the greater engagement in exercise itself, but rather could be attributable to a range of other benefits associated with the programme. While the

extent to which our findings can be generalised to other exercise (or SCT-based behavioural coaching) programmes for PwMS is unclear, and potentially programme dependent, these findings do suggest that similarly structured programmes should be encouraged and investigated for psychosocially beneficial outcomes.

Interestingly, PwMS had significantly reduced STAI scores following completion of the MoveSmart programme. In addition, 20.4% of participants scored above the cut-off for clinically significant anxiety at baseline, with 6.25% of participants scoring above that same cut-off post-programme. While we would suggest against pathologising the experience of anxiety in PwMS, reductions in clinically significant levels of anxiety can be taken as encouraging given the relationship between clinically significant anxiety and unwanted symptomology such as higher depression and fatigue in PwMS [52]. Despite trait anxiety theoretically being more stable over time than state anxiety, results of this study suggest that it is also amenable to change and align with previous findings in this area [53–55]. Given the association between trait anxiety and a number of mood disorders, programmes which can effectively reduce trait anxiety may be of particular value for PwMS [56,57]. In addition, improvements in a range of symptomology beyond anxiety were reported following the programme, including improvements in exercise habits and fatigue, as well as reductions in the reported physical and psychological impact of MS.

A number of sociodemographic and health-related factors were associated with anxiety at baseline, which is consistent with previous research [58,59]. Notably, the finding of an association between younger age and higher anxiety has been previously reported in MS populations [59] and suggests that PwMS of a younger age may be more likely to experience anxiety. However, it is worth noting that findings surrounding the relationship between age and anxiety in PwMS vary, with several studies investigating this relationship reporting findings of non-significance [6,59].

It is interesting to note that higher anxiety at baseline was associated with lower disability (as measured by the PDDS), but higher perceived physical impact from MS (as measured by the MSIS-29). However, while these findings may appear contradictory, it is important to note that these two measures capture different aspects of experience among PwMS. While the PDDS specifically measures one aspect of disability, specifically mobility disability, the MSIS physical subscale captures a range of perceived impacts of MS, beyond mobility impacts. The MSIS-29 physical subscale is not an objective measure of the sum of these physical issues but instead measures the extent to which these issues impact on an individual's quality of life [41]. The experience of MS symptoms can vary greatly with some individuals experiencing low disability in terms of mobility, but a number of other physical issues which impact daily functioning (e.g., neurogenic bladder) [60,61]. Thus, our findings suggest that, while those with less mobility disability may be at risk of experiencing higher anxiety, identifying additional supports and help with coping strategies to lessen impact of physical MS symptomology on quality of life may be the most beneficial way to help alleviate anxious symptomology.

This study also includes a separate measure of fatigue, which is one of the most common and impactful symptoms associated with MS [58,62]. It is perhaps unsurprising therefore that fatigue at baseline independently predicted pre-programme anxiety, aligning with previous findings of an association between anxiety and fatigue in PwMS [52,58]. While significant associations have previously been found between disability and fatigue [63], when accounting for level of disability fatigue has been found to independently predict quality of life outcomes in PwMS [64–66]. However, previous research has reported considerable variance in the strength of associations between fatigue and anxiety [67]. The directionality of the relationship between fatigue and anxiety remains unclear [65], however there is considerable suggestion that fatigue and anxiety may have a bi-directional relationship, suggesting potential efficacy for anxiety supports to impact fatigue symptoms in PwMS [68,69].

Despite the MoveSmart programme successfully and significantly improving anxiety and exercise habits (the target variable of the programme), changes in exercise habits had no significant association with change in anxiety, suggesting that improvements in anxiety were due to other factors. Indeed, change in perceived physical impacts of MS following participation in the MoveSmart programme was the only significant predictor of improvements in anxiety which indicates that this may be a key driver of the observed change. This finding highlights the importance of addressing the impact of

MS symptoms on quality of life, with associations (and a potentially bi-directional relationship) between quality of life and anxiety well established [70]. This finding also highlights the importance of the physical symptoms of MS in influencing the experience of anxiety in PwMS. However, while changes in the physical impact of MS on quality of life predicted a significant percentage of the variance in change in anxiety, it is possible that other factors not included in our model also contributed to this change. For example, while social benefits were described in the qualitative data, mirroring findings from Russell et al [30], there was no quantitative measure which captured level of social support in the pre- and post- programme surveys. It is known that social support has strong associations with anxiety [4,6], and that PwMS have a desire for peer support, particularly for supporting the management of anxious symptomology [30]. It is possible that the peer support experienced as part of the programme was a key factor in improving psychological wellbeing.

Focus group findings of considerable praise for programme execution align with previous research [32] in which participants praised the execution of services provided by MS Ireland. While this research highlighted the considerable desire for peer supports among the MS population, descriptions of the value of peer support received during the MoveSmart programme further highlight the importance of the availability of these supports for PwMS.

Another interesting finding from this analysis is that, while improvements in self-efficacy were noted by focus-group participants, exercise self-efficacy scores did not significantly change following completion of the programme. These findings mirror those of Coote et al [29], where improvements for anxiety were reported following completion of the SCT version of the Step it Up programme, with only small, non-significant improvements reported for exercise self-efficacy. One possible explanation for this is that while some participants saw expected self-efficacy improvements, other participants may have overestimated their exercise abilities prior to engaging in MoveSmart programme, gaining a more realistic understanding of their current abilities through engaging in additional exercise. Additionally, it is feasible that some participants experienced benefits related to general self-efficacy, with SCT based behavioural coaching aimed at fostering this improvement, but did not improve in exercise self-efficacy which is a more specific construct. Additionally, while the EXSE has been commonly used and validated in MS populations it is possible that this finding may have been influenced by limitations with the measure itself, with suggestions that a Rasch analysis may be used to further investigate and improve the validity of this scale [71].

Qualitative findings surrounding the benefits of flexible goal adjustment are supported by general population studies showing associations between goal disengagement, goal reengagement and higher quality of life [72]. The experienced benefit from both goal setting and routine creating elements of MoveSmart are echoed by qualitative evaluations of the Step it Up programmes [30]. Similarly, findings from the Step It Up programme suggest that higher exercise goal setting is associated with higher levels of PA, which may suggest the potential for exercise goal setting to influence anxiety through PA. MS research involving concepts of goal setting and goal adjustment remains limited, with considerable need for further research into the mechanisms underlying potential associations and support pathways, but limited research in this area suggests a potentially significant relationship between anxiety and goal related behaviours [6,73,74].

## 4.1 Strengths and limitations

To our knowledge, this is the first study which included anxiety as a primary outcome for assessing an exercise intervention for PwMS [13] The longitudinal design of this study allows for comparison of factors at multiple time points. Other strengths of this study include its large sample size and real-world pragmatic approach to evaluation of a support service being used in a clinical setting. However, while this approach has many benefits, including allowing for direct recommendations for clinical care, this approach does not allow for the same level of control as an RCT. Additionally patient-reported measures, used here for feasibility reasons, are subject to a number of biases which may have impacted results. For example, evidence suggests that self- reported physical activity may only be weakly correlated with objective physical activity in PwMS [75]. Similarly, PDDS scores have recently been suggested to have only weak correlations with the Expanded Disability Status Scale (EDSS), a neurologist-led assessment of disability [76]. While a considerable number

of participants did not complete post-programme measures, our analysis found no significant differences in completers vs. non-completers on validated measures. Despite the findings from these sensitivity analyses, it is possible that the attrition rate (54%) may have introduced potential response biases or impacted the internal validity of our findings. For the purpose of this analysis, scale variables from validated measures were treated as continuous variables. While this is commonplace [77–81] in social sciences research, this approach limits the specificity with which we can interpret our findings given the potential for inequitable difference between points on these scales [82,83]. Data was taken from participants who have engaged with an exercise programme and therefore may not be fully representative of all PwMS. As some data collection may have taken place during and immediately following the COVID-19 pandemic, it is unclear if this context impacted on the findings presented here. While there may have been improvements in anxiety and other outcomes immediately after the programme, it is unclear the extent to which these improvements were sustained in the time following completion of post-programme measures. Data on race or ethnicity were not available for this sample, so it is unclear whether our sample was representative in this respect. We did not have a record of if participants were taking any medication for anxiety or depression which may have impacted on our results. While the STAI-T remains a commonly used measure of trait anxiety in both general and MS populations [84,85], there is some suggestion that the STAI-T may be a better measure of non-specific negative affect than trait anxiety, with further suggestion that measures should be developed which can more accurately assess individual's typical threat responses [86].

### 4.2 Conclusion

The MoveSmart MS programme shows good efficacy in improving several concepts related to physical and mental wellbeing in PwMS. Quantitative analysis of the mechanisms through which the programme achieved improvements in participant trait anxiety suggest that reductions in the impact of physical symptoms on quality of life may play a key role in this outcome. Additionally, qualitative data collected from focus groups suggest that peer support and concepts related to goal setting and flexible goal adjustment were seen as particularly valuable by programme participants. Additionally, several concepts associated with anxiety at baseline were identified, including younger age, lower mobility disability, higher fatigue and higher physical impact of MS on quality of life, which may help with the identification of individuals who may have a need for psychological support. Future research should continue to explore the associates of anxiety as well as the mechanisms driving anxiety improvements, with a view to improving existing supports and informing the development of future supports for PwMS.

### Supporting information

**S1 File. Human participants research checklist.**
(PDF)

**S2 File. Chair's decision research ethics application form.**
(PDF)

### Acknowledgments

We would like to thank MS Ireland for their help in facilitating this study.

### Author contributions

**Conceptualization:** Austin Fahy, Susan Coote, Rebecca Maguire.

**Data curation:** Austin Fahy.

**Formal analysis:** Austin Fahy.

**Investigation:** Austin Fahy.

**Methodology:** Austin Fahy, Rebecca Maguire.

**Resources:** Susan Coote.

**Supervision:** Rebecca Maguire.

**Writing – original draft:** Austin Fahy.

**Writing – review & editing:** Austin Fahy, Susan Coote, Rebecca Maguire.

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
