## [Decision Letter · Decision Letter 0]

13 Jul 2025

Dear Dr. Maguire,

Thank you for submitting your manuscript to PLOS ONE. After careful consideration, we feel that it has merit but does not fully meet PLOS ONE’s publication criteria as it currently stands. Therefore, we invite you to submit a revised version of the manuscript that addresses the points raised during the review process.

We look forward to receiving your revised manuscript.

Kind regards,

Markus Reindl

Academic Editor

PLOS ONE

Journal Requirements:

Reviewers' comments:

Reviewer's Responses to Questions

**Comments to the Author**

1. Is the manuscript technically sound, and do the data support the conclusions?

Reviewer #1: Yes

Reviewer #2: Yes

2. Has the statistical analysis been performed appropriately and rigorously?

Reviewer #1: No

Reviewer #2: No

3. Have the authors made all data underlying the findings in their manuscript fully available?

Reviewer #1: Yes

Reviewer #2: Yes

4. Is the manuscript presented in an intelligible fashion and written in standard English?

Reviewer #1: Yes

Reviewer #2: Yes

Reviewer #1: PONE-D-25-06696

Evaluation of MoveSmart MS, an online structured exercise SCT-based behavioural coaching and peer support programme, on anxiety in multiple sclerosis

Reviewer’s report

Dear authors,

Your study addresses a relevant and novel topic, is clearly structured, and well written. Please find below my suggestions aimed at enhancing the overall quality and clarity of your manuscript.

Title and keywords

I assume that many potential readers may not be familiar with the abbreviation ‘SCT’ and could interpret it as referring to ‘stem cell transplant’. To ensure the title attracts interest and is easily understood, I recommend revising it to include the full term instead of the abbreviation. This also applies to the keywords, where you could use the MeSH term ‘Psychological Theory’ or the entry term ‘Social Cognitive Theory’.

Abstract

In the abstract, it is stated that ‘younger age, lower disability, higher physical impact of MS and higher fatigue were significantly associated with higher baseline anxiety scores, and changes in the physical impact of MS predicted changes in anxiety scores.’ As lower disability is typically associated with lower physical impact, this may appear contradictory to readers. It seems likely that "physical impact" refers to a specific measurement, possibly the physical impact subscale of the MSIS-29. If this is the case, I suggest making this explicit in the abstract to ensure clarity.

Introduction

Lines 120-125: The sentence discussing the role of self-efficacy in relation to anxiety via negative problem orientation is conceptually strong. However, I recommend clarifying that self-efficacy is a positive construct, and it is the level of self-efficacy (i.e. high vs low) that is associated with differences in negative problem orientation. A brief elaboration of this pathway could help readers unfamiliar with these concepts better understand the proposed mechanism.

Lines 177-182: Please ensure consistency in the use of tense when stating the study aims. I recommend using past tense, which is generally preferred when referring to the aims of a completed study.

Method

Line 193: Please ensure that British English is used consistently throughout the manuscript—for example, replace behavioral with behavioural.

Line 209: Please include the name of the research ethics committee and reference number.

Line 299: Kindly add the missing full stop at the end of the sentence.

Data analysis: The statistical methods applied (e.g., t-tests, hierarchical regression, and the use of means and standard deviations) may not be entirely appropriate, given that a large proportion of the data appear to be ordinal in nature [1]. I recommend that the authors clarify which variables are ordinal and justify their treatment as continuous where applicable. If parametric analyses are to be used, one option is to apply ordinal-to-interval transformation tables, such as those derived from Rasch analysis, if available for the instruments used [2]. Alternatively, the authors could consider using non-parametric tests (e.g., the Wilcoxon signed-rank test) for comparisons and ordinal logistic regression for modelling outcomes. In addition, for purely ordinal data, reporting medians and interquartile ranges (IQR) may be more appropriate than means and standard deviations.

Results

I suggest using more meaningful and descriptive subheadings to guide the reader through the findings.

In addition, it would be preferable to describe all statistical analyses in the relevant Methods section and remove redundant explanations from the Results section—for example, on lines 362–363, 374–375, and 394. This would streamline the Results and help maintain a clearer separation between methods and findings.

Table 1: The reporting of only male and female categories suggests a binary understanding of gender. If the variable refers to biological characteristics, please clarify this and consider changing ‘gender’ to ‘sex.’

Table 2: In line with my concern regarding the statistical analysis, I would argue that questionnaire data based on Likert scales are not continuous in nature. Therefore, the table heading should be revised accordingly, and the presentation of the data should be explicitly justified. Furthermore, kindly specify all abbreviations used in the table within the legend; this refers to all tables.

Lines 401-407: Age and disease duration are typically correlated, which may lead to multicollinearity when both are included in regression models. Was multicollinearity between these variables assessed in this sample? If so, please report the relevant statistics (e.g., correlation coefficient, VIF) and clarify whether it posed any concern for the analyses.

Table 4: Please correct the typographical error in the table legend by adding a ‘t’ to ‘limi’, so it reads “limit”. For the variable ‘time since diagnosis’, please use lowercase letters within the brackets—for example: [1 = 5 years or less, 2 = more than 5 years]—to ensure consistency in formatting; the same applies to Table 6.

Chapter 3.5

Kindly revise the term ‘program’ to ‘programme’ throughout the manuscript to ensure consistency with British English spelling.

Table 7 / Qualitative findings:

The verbatim quotes effectively illustrate the participants’ experiences. Would it be possible to replace some of the quotes that are cited more than once with additional quotes from other participants? This would reduce redundancy and potentially enrich the insights presented to the reader.

Discussion

The authors state that ‘another interesting finding from this analysis is that, while improvements in self-efficacy were noted by focus-group participants, exercise self-efficacy scores did not significantly change following completion of the programme.’ Could the lack of improvement be related to limitations of the EXSE itself? While the scale has been used in previous MS studies, the items appear somewhat overlapping, and to my knowledge, it has not yet been subjected to Rasch analysis in people with MS. I recommend that the authors briefly reflect on this possibility in the Discussion section.

References

1. Verhulst B, Neale MC. Best Practices for Binary and Ordinal Data Analyses. Behav Genet. 2021;51(3):204-14. Epub 2021/01/06. doi: 10.1007/s10519-020-10031-x. PubMed PMID: 33400061; PubMed Central PMCID: PMCPMC8096648.

2. Vanhoutte EK, Hermans MC, Faber CG, Gorson KC, Merkies IS, Thonnard JL. Rasch-ionale for neurologists. J Peripher Nerv Syst. 2015;20(3):260-8. Epub 2015/06/27. doi: 10.1111/jns.12122. PubMed PMID: 26115370.

Reviewer #2: General Comments

This is a well-conducted study addressing an important and underexplored topic in the context of multiple sclerosis. The paper provides a substantial amount of detail and incorporates both quantitative and qualitative data.

Title & Abstract

The full form of SCT should be written out in the title.

The design of the study should be clearly stated either in the title or in the abstract.

Introduction

Please don’t use an abbreviation (MS) before it has been defined.

The section introducing the MoveSmart programme is informative but would benefit from clearer linkage to the preceding background. Although the programme is positioned as building on prior work, the specific ways in which the current study differs conceptually or methodologically remain unclear.

While the introduction draws on a wide range of literature, the engagement is largely descriptive. A more critical synthesis of past findings would strengthen the rationale for the current study.

Methods

The following sentence requires a supporting reference: “This study involves secondary analysis of data from participants who took part in the MoveSmart MS programme.”

While the study is described as a secondary analysis, it is not sufficiently clear what the original purpose of data collection was. Further detail is needed to clarify how the current analysis is distinct from or extends the original purpose.

The description of the MoveSmart intervention is insufficiently detailed.

The qualitative procedures lack transparency. Additional information is required on sampling strategy, number of participants per group, duration of each focus group session, etc.

The rationale for excluding QIDS and MSIS-29 psychological subscale scores due to multicollinearity is reasonable; however, the statistical method used to detect multicollinearity (e.g. VIF, tolerance) should be specified.

Power analysis is mentioned only briefly. The assumptions underlying the calculation—including effect size, alpha level, power level, and expected number of predictors—should be reported.

Results

Clarify how many participants completed both pre- and post-programme assessments.

The use of Santangelo et al. (2016) to adjust STAI cut-offs should be justified in the Methods section, not just mentioned in the Results.

In Table 2, p-values alone are insufficient. Consider including effect sizes (e.g. Cohen’s d) or referring to Minimal Clinically Important Differences (MCID) where available to support interpretation of meaningful change.

For multicollinearity, the cut-off criteria used for exclusion should be explicitly reported in the statistical analysis section. Instead of relying solely on correlation coefficients (r), multicollinearity should be assessed using tolerance and VIF values.

Discussion

The clinical significance of the observed reduction in anxiety is not sufficiently contextualised.

Although goal setting is presented as beneficial, the underlying mechanisms are not adequately elaborated.

The high attrition rate (54%) is not critically discussed. It may have implications for internal validity and potential response bias.

The generalizability of the findings is not addressed.

The Ireland-specific context is prominent in the Introduction but not revisited in the Discussion. It is unclear why this geographical framing is emphasised and whether it has relevance for service delivery, healthcare access, or cultural context.

**Do you want your identity to be public for this peer review?** For information about this choice, including consent withdrawal, please see our Privacy Policy

Reviewer #1: No

Reviewer #2: No

---

## [Author Response · Author response to Decision Letter 1]

30 Sep 2025

Many thanks for your helpful feedback on our manuscript. Please find our responses below.

Reviewer 1 comments:

Dear authors,

Your study addresses a relevant and novel topic, is clearly structured, and well written. Please find below my suggestions aimed at enhancing the overall quality and clarity of your manuscript.

Response: Thank you for your kind and constructive feedback.

Title and keywords

I assume that many potential readers may not be familiar with the abbreviation ‘SCT’ and could interpret it as referring to ‘stem cell transplant’. To ensure the title attracts interest and is easily understood, I recommend revising it to include the full term instead of the abbreviation. This also applies to the keywords, where you could use the MeSH term ‘Psychological Theory’ or the entry term ‘Social Cognitive Theory’.

Response: Thank you for pointing this out. The title has now been revised to refer to the full term “Social Cognitive Theory”. We have also made the suggested change to our keywords.

Abstract

In the abstract, it is stated that ‘younger age, lower disability, higher physical impact of MS and higher fatigue were significantly associated with higher baseline anxiety scores, and changes in the physical impact of MS predicted changes in anxiety scores.’ As lower disability is typically associated with lower physical impact, this may appear contradictory to readers. It seems likely that "physical impact" refers to a specific measurement, possibly the physical impact subscale of the MSIS-29. If this is the case, I suggest making this explicit in the abstract to ensure clarity.

Response: Thank you for this suggestion, you are correct in your assumption. We have replaced “physical impact” with “MSIS-29 physical scores” as recommended to improve clarity.

Introduction

Lines 120-125: The sentence discussing the role of self-efficacy in relation to anxiety via negative problem orientation is conceptually strong. However, I recommend clarifying that self-efficacy is a positive construct, and it is the level of self-efficacy (i.e. high vs low) that is associated with differences in negative problem orientation. A brief elaboration of this pathway could help readers unfamiliar with these concepts better understand the proposed mechanism.

Response: Thank you for this suggestion. We have now provided additional clarification on the nature of self-efficacy and the directionality of its relationship with negative problem orientation.

Lines 177-182: Please ensure consistency in the use of tense when stating the study aims. I recommend using past tense, which is generally preferred when referring to the aims of a completed study.

Response: Thank you for picking up on this inconsistency. We have rectified this now.

Method

Line 193: Please ensure that British English is used consistently throughout the manuscript—for example, replace behavioral with behavioural.

Response: This has now been addressed here and throughout the manuscript.

Line 209: Please include the name of the research ethics committee and reference number.

Response: The name of the research ethics committee (University of Limerick ethics committee) and reference no. (2020-12-19) has now been added to this section.

Line 299: Kindly add the missing full stop at the end of the sentence.

Response: Thank you for finding this typo, we have addressed this now.

Data analysis: The statistical methods applied (e.g., t-tests, hierarchical regression, and the use of means and standard deviations) may not be entirely appropriate, given that a large proportion of the data appear to be ordinal in nature [1]. I recommend that the authors clarify which variables are ordinal and justify their treatment as continuous where applicable. If parametric analyses are to be used, one option is to apply ordinal-to-interval transformation tables, such as those derived from Rasch analysis, if available for the instruments used [2]. Alternatively, the authors could consider using non-parametric tests (e.g., the Wilcoxon signed-rank test) for comparisons and ordinal logistic regression for modelling outcomes. In addition, for purely ordinal data, reporting medians and interquartile ranges (IQR) may be more appropriate than means and standard deviations.

Response: Thank you for these observations. While scaled values from validated measures are commonly treated as continuous variables in social sciences research (e.g. Chiou et al. 2023, Huang et al., 2021, Miki et al., 2023, Lacomba‐Trejo et al., 2022, Yang et al., 2025 etc.), we acknowledge that there are some technical limitations to this approach which limit the specificity with which we can describe the nature of the changes and relationships described here. We have now explicitly acknowledged these limitations in section 4.1. In the context of the aims of this study, the prevalence of this approach and the fact that our data met assumptions of normality (either being binary coded or based on validated scales), we are satisfied that the analytic approach taken here is appropriate.

Results

I suggest using more meaningful and descriptive subheadings to guide the reader through the findings.

Response: Upon reflection we agree that our previously provided headings in this section lack depth. We have now updated these headings to include more detailed descriptions of their corresponding subsections.

In addition, it would be preferable to describe all statistical analyses in the relevant Methods section and remove redundant explanations from the Results section—for example, on lines 362–363, 374–375, and 394. This would streamline the Results and help maintain a clearer separation between methods and findings.

Response: We have removed redundant explanations to improve the distinction between these sections.

Table 1: The reporting of only male and female categories suggests a binary understanding of gender. If the variable refers to biological characteristics, please clarify this and consider changing ‘gender’ to ‘sex.’

Response: Thank you for pointing this out. This has now been corrected.

Table 2: In line with my concern regarding the statistical analysis, I would argue that questionnaire data based on Likert scales are not continuous in nature. Therefore, the table heading should be revised accordingly, and the presentation of the data should be explicitly justified. Furthermore, kindly specify all abbreviations used in the table within the legend; this refers to all tables.

Response: Thank you for these considerations. We have decided to simplify the Table 2 heading accordingly to avoid confusion regarding the nature of our variables. We discuss our treatment of these variables in greater detail above and also now in section 4.1. We have also ensured that all abbreviations are now specified in the table legend.

Lines 401-407: Age and disease duration are typically correlated, which may lead to multicollinearity when both are included in regression models. Was multicollinearity between these variables assessed in this sample? If so, please report the relevant statistics (e.g., correlation coefficient, VIF) and clarify whether it posed any concern for the analyses.

Response: The reviewer raises a good point here. We also considered this a possibility but can confirm that having examined VIF and tolerance values there was no significant multicollinearity between age and time since diagnosis in this sample. While time since diagnosis and age can often be correlated, they are distinct, given the considerable variance in age of diagnosis that PwMS may experience.

Table 4: Please correct the typographical error in the table legend by adding a ‘t’ to ‘limi’, so it reads “limit”. For the variable ‘time since diagnosis’, please use lowercase letters within the brackets—for example: [1 = 5 years or less, 2 = more than 5 years]—to ensure consistency in formatting; the same applies to Table 6.

Response: These changes have been made.

Chapter 3.5

Kindly revise the term ‘program’ to ‘programme’ throughout the manuscript to ensure consistency with British English spelling.

Response: British English spelling has been consistently applied throughout the document.

Table 7 / Qualitative findings:

The verbatim quotes effectively illustrate the participants’ experiences. Would it be possible to replace some of the quotes that are cited more than once with additional quotes from other participants? This would reduce redundancy and potentially enrich the insights presented to the reader.

Response: We agree and have ensured all quotes feature only once to maximize the richness of the data presented.

Discussion

The authors state that ‘another interesting finding from this analysis is that, while improvements in self-efficacy were noted by focus-group participants, exercise self-efficacy scores did not significantly change following completion of the programme.’ Could the lack of improvement be related to limitations of the EXSE itself? While the scale has been used in previous MS studies, the items appear somewhat overlapping, and to my knowledge, it has not yet been subjected to Rasch analysis in people with MS. I recommend that the authors briefly reflect on this possibility in the Discussion section.

Response: Thank you for this suggestion. We had our own reservations about the limitations of the EXSE and are happy to suggest that a Rasch analysis would be valuable in illustrating potential limitations as well as facilitating iteration. We have now added this suggestion to section 4.1.

References

1. Verhulst B, Neale MC. Best Practices for Binary and Ordinal Data Analyses. Behav Genet. 2021;51(3):204-14. Epub 2021/01/06. doi: 10.1007/s10519-020-10031-x. PubMed PMID: 33400061; PubMed Central PMCID: PMCPMC8096648.

2. Vanhoutte EK, Hermans MC, Faber CG, Gorson KC, Merkies IS, Thonnard JL. Rasch-ionale for neurologists. J Peripher Nerv Syst. 2015;20(3):260-8. Epub 2015/06/27. doi: 10.1111/jns.12122. PubMed PMID: 26115370.

Reviewer 2 comments:

This is a well-conducted study addressing an important and underexplored topic in the context of multiple sclerosis. The paper provides a substantial amount of detail and incorporates both quantitative and qualitative data.

Response: Thank you for your kind feedback.

Title & Abstract

The full form of SCT should be written out in the title.

Response: This change has been made.

The design of the study should be clearly stated either in the title or in the abstract.

Response: The study design (pre/post evaluation) has now been included in the abstract.

Introduction

Please don’t use an abbreviation (MS) before it has been defined.

Response: Apologies for this oversight. This abbreviation has now been defined in full on first use.

The section introducing the MoveSmart programme is informative but would benefit from clearer linkage to the preceding background. Although the programme is positioned as building on prior work, the specific ways in which the current study differs conceptually or methodologically remain unclear.

Response: We have now explained that MoveSmart builds upon the prior Step It Up programme by including MS-symptom specific modules alongside the core modules of Step It Up. We have provided additional detail and exemplars both in the Introduction and in the Methods section.

While the introduction draws on a wide range of literature, the engagement is largely descriptive. A more critical synthesis of past findings would strengthen the rationale for the current study.

Response: Thank you for this suggestion. We have revisited this section and added some additional rationale and context for this study.

Methods

The following sentence requires a supporting reference: “This study involves secondary analysis of data from participants who took part in the MoveSmart MS programme.”

While the study is described as a secondary analysis, it is not sufficiently clear what the original purpose of data collection was. Further detail is needed to clarify how the current analysis is distinct from or extends the original purpose.

Response: Reference to secondary analysis has been removed. After some consideration we agree that this analysis aligns with the original purpose of the data collection, which was evaluating the MoveSmart MS programme.

The description of the MoveSmart intervention is insufficiently detailed.

Response: We have provided additional detail describing the MoveSmart programme, including module exemplars, how this is an iteration from the Step It Up programme, and some more specifics of the programme delivery.

The qualitative procedures lack transparency. Additional information is required on sampling strategy, number of participants per group, duration of each focus group session, etc.

Response: We have now provided additional information and clarification regarding sampling, focus-group duration and the number of focus group sessions (n=1) which should help improve clarity.

The rationale for excluding QIDS and MSIS-29 psychological subscale scores due to multicollinearity is reasonable; however, the statistical method used to detect multicollinearity (e.g. VIF, tolerance) should be specified.

Response: Thank you for highlighting this oversight. We now explicitly confirm in the Results that VIF and tolerance values were consulted in our assessment of multicollinearity. In addition to statistical concerns regarding multicollinearity, these excluded variables were excluded due to the conceptual overlap they share with our the outcome variable anxiety.

Power analysis is mentioned only briefly. The assumptions underlying the calculation—including effect size, alpha level, power level, and expected number of predictors—should be reported.

Response: We have now reported the assumptions underlying this power analysis.

Results

Clarify how many participants completed both pre- and post-programme assessments.

Response: We have added an extra line to clarify that 160 participants completed both pre and post-programme assessments.

The use of Santangelo et al. (2016) to adjust STAI cut-offs should be justified in the Methods section, not just mentioned in the Results.

Response: We have now provided justification for the use of clinical cut-offs in accordance with Santangelo et al. (2016) in the Methods section.

In Table 2, p-values alone are insufficient. Consider including effect sizes (e.g. Cohen’s d) or referring to Minimal Clinically Important Differences (MCID) where available to support interpretation of meaningful change.

Response: Thank you for suggestion. We agree that providing this information would help to contextualise these findings. We have now added a column detailing effect sizes in Table 2.

For multicollinearity, the cut-off criteria used for exclusion should be explicitly reported in the statistical analysis section. Instead of relying solely on correlation coefficients (r), multicollinearity should be assessed using tolerance and VIF values.

Response: Thank you for highlighting this. We can confirm that both VIF and tolerance values were consulted in our assessment of multicollinearity. We now explicitly state that VIFs greater than 4 and tolerance values less than 0.25 were considered indicative of multicollinearity.

Discussion

The clinical significance of the observed reduction in anxiety is not sufficiently contextualised.

Response: We have provided additional context for the investigation and findings related to clinically significant anxiety as well as clarification of the value of these findings.

Although goal setting is presented as beneficial, the underlying mechanisms are not adequately elaborated.

Response: We agree that exploration of these mechanisms would be valuable. While these explorations were outside the scope of what was feasible in this study, there is a need for greater understanding in this area particularly for PwMS which we have made clear in our call for future research.

The high attrition rate (54%) is not critically discussed. It may have implications for internal validity and potential response bias.

Response: We have now more explicitly acknowledged the potential impacts of o

---

## [Decision Letter · Decision Letter 1]

20 Oct 2025

Dear Dr. Maguire,

We look forward to receiving your revised manuscript.

Kind regards,

Markus Reindl

Academic Editor

PLOS ONE

Journal Requirements:

Reviewers' comments:

Reviewer's Responses to Questions

**Comments to the Author**

Reviewer #1: All comments have been addressed

Reviewer #2: All comments have been addressed

2. Is the manuscript technically sound, and do the data support the conclusions?

Reviewer #1: Yes

Reviewer #2: Yes

3. Has the statistical analysis been performed appropriately and rigorously?

Reviewer #1: Yes

Reviewer #2: Yes

4. Have the authors made all data underlying the findings in their manuscript fully available?

Reviewer #1: Yes

Reviewer #2: No

5. Is the manuscript presented in an intelligible fashion and written in standard English?

Reviewer #1: Yes

Reviewer #2: Yes

Reviewer #1: PONE-D-25-06696R1

Evaluation of MoveSmart MS, an online structured exercise SCT-based behavioural coaching and peer support programme, on anxiety in multiple sclerosis

Reviewer’s report

Dear Authors,

Thank you for the revisions made to your manuscript.

There are just a few minor points I would like you to address:

Abstract: Please specify the abbreviation MSIS-29 at its first occurrence.

Tables 2–6 (legends): Kindly list the abbreviations in alphabetical order.

Table 7, Subtheme ‘Continued meet-ups encouraged continued exercise habit’: Please begin the quotation with a capital letter.

Reviewer #2: The authors have revised the manuscript well and addressed my concerns. I have no additional comments.

**Do you want your identity to be public for this peer review?** For information about this choice, including consent withdrawal, please see our Privacy Policy

Reviewer #1: No

Reviewer #2: No

---

## [Author Response · Author response to Decision Letter 2]

25 Oct 2025

Thank you for your feedback on the latest version of our submitted manuscript entitled “Evaluation of MoveSmart MS, an online structured exercise, Social Cognitive Theory-based behavioural coaching and peer support programme on anxiety in multiple sclerosis”.

Please find enclosed our updated and amended submission [PONE-D-25-06696R2]. These updates have been made in response to the minor recommendations outlined by the reviewers which we have included below.

Response to Comments:

Journal Requirements:

Response: Reviewer comments did not include recommendations for citations.

Response: We have checked and ensured our reference list is complete and accurate. There were no changes to references or citations in these revisions.

Reviewer #1’s comments:

Dear Authors,

Thank you for the revisions made to your manuscript.

There are just a few minor points I would like you to address:

1. Abstract: Please specify the abbreviation MSIS-29 at its first occurrence.

Response: We have now included the unabbreviated name of this scale (Multiple Sclerosis Impact Scape-29) in the abstract.

2. Tables 2–6 (legends): Kindly list the abbreviations in alphabetical order.

Response: We know have alphabetised our abbreviations lists on these tables.

3. Table 7, Subtheme ‘Continued meet-ups encouraged continued exercise habit’: Please begin the quotation with a capital letter.

Response: We have now corrected this as suggested.

Reviewer #2’s comments:

The authors have revised the manuscript well and addressed my concerns. I have no additional comments.

Response: Thank you for the positive feedback.

We would like to express our gratitude again towards the editor and both reviewers of this submission for helping us to improve our manuscript.

---

## [Editor Report · Decision Letter 2]

27 Oct 2025

Evaluation of MoveSmart MS, an online structured exercise, Social Cognitive Theory-based behavioural coaching and peer support programme, on anxiety in multiple sclerosis

PONE-D-25-06696R2

Dear Dr. Maguire,

We’re pleased to inform you that your manuscript has been judged scientifically suitable for publication and will be formally accepted for publication once it meets all outstanding technical requirements.

Kind regards,

Markus Reindl

Academic Editor

PLOS ONE
---

## [Editor Report · Acceptance letter]

PONE-D-25-06696R2

PLOS ONE

Dear Dr. Maguire,

I'm pleased to inform you that your manuscript has been deemed suitable for publication in PLOS ONE. Congratulations! Your manuscript is now being handed over to our production team.

Kind regards,

on behalf of

Dr. Markus Reindl

Academic Editor

PLOS ONE